# The Inclusion of Other-Sex Peers in Peer Networks and Sense of Peer Integration in Early Adolescence: A Two-Wave Longitudinal Study

**DOI:** 10.3390/ijerph192214971

**Published:** 2022-11-14

**Authors:** Paweł Grygiel, Sławomir Rębisz, Anna Gaweł, Barbara Ostafińska-Molik, Małgorzata Michel, Julia Łosiak-Pilch, Roman Dolata

**Affiliations:** 1Institute of Education, Jagiellonian University, 31-135 Cracow, Poland; 2Institute of Education, Rzeszów University; 35-010 Rzeszow, Poland; 3Faculty of Education, University of Warsaw, 00-561 Warsaw, Poland

**Keywords:** peer relations, sex heterophily, sex homophily, sociometric popularity, early adolescents, well-being

## Abstract

The main goal of the analysis presented in this paper is to examine the dynamics of including other-sex peers in the peer networks of early adolescents, aged 11 (at T1) and 13 (at T2), and the relationship between sex heterophily and changes in the sense of peer integration. The analysis was conducted using the Latent Difference Score (LDS) model with data from a representative nationwide longitudinal study in Poland (n = 5748). With reference to the dynamics related to the heterophilic process, the research confirmed that at the beginning of grade 5 of primary school, heterophily is still relatively rare, yet towards the end of early adolescence, there is a gradual shift, more strongly in girls, towards breaking through the strictly same-sex segregation and embarking on heterophilic relationships. Importantly, the LDS model—even when controlling for different measures of peer network—showed significant and positive (among both girls and boys) relations between establishing cross-sex relationships and the sense of peer integration. The results indicate that the appearance of the opposite sex in the peer network between grades 5 and 6 will improve the sense of peer integration. The findings are discussed in relation to results from other studies in the field.

## 1. Introduction

During early adolescence (age 11–14), peers become an increasingly important source of instrumental, social, and emotional support [1], and adolescents are more susceptible to the influence of peers than at any other time during the course of their lives. This is also the result of compulsory education. For example, in the USA, students spend on average 170–180 days a year, five days a week, six and a half hours a day at school [2,3]. Similarly, in Poland, it takes almost 6000 h to take a student through primary education, and this figure only includes compulsory lessons [4]. Being outside of the home environment, the consequently diminished role of parents and increased significance of peers [5] result in the school environment becoming a space of not only intellectual but also mental and social development in the period of adolescence [6]. An important area of social development is gender relations.

### 1.1. Homophily Based on Sex Differences

Early adolescence is characterized mostly by relationships with peers occurring in same-sex groups. Sex-based homophily occurs as early as between the 18th and 28th month of human life [7], becoming distinct between the 30th and 36th month and peaking at 11 years of age [8]. From the age of 12, the intensity of this process decreases, although the tendency to form new relationships with people of the same sex rather than the opposite sex persists throughout primary school [9,10,11,12,13,14,15,16,17]. Low attraction of opposite sex peers during this period is associated with various leisure activities [18], a preference for different types of play [19], reading different literary genres [20], and a preference for different music styles [21].

The relational aspect of sex homophily is accompanied by the specific structure of girls’ and boys’ peer networks and the different functions they play [22]. It has been demonstrated that girls’ peer networks are characterized by more horizontal and intimate relationships, whereas boys’ networks are broader and more hierarchical [23]. Close relationships with others are valuable to all adolescents, yet for girls, it is emotional support that is a source of closeness, while for boys, it comes from shared activities [23].

Barriers to gender relations are largely created at the group level and internalized at the individual level [24]. Consequently, adolescents who establish relationships with persons of the other sex can be exposed to sanctions, e.g., stigmatization. Girls are sometimes described as “promiscuous” and boys as “gay” [12]. Research shows that children who show a tendency towards sex-atypical behavior are vulnerable to peer sanctions. Boys feel more pressure to conform to gender norms [25,26], and at the same time are more pressurized to conform [27,28].

At the pre-pubertal stage, other-sex peer relationships are relatively rare, but sex homophily is not complete. For example, in the USA, about 25% of adolescents’ friends tend to be cross-sex peers [29]. In Poland, research indicates that at the age of 11–12—and thus at the peak of the intensity of sex-based homophily—around 10% of pupils also identify people of the other sex as liked [30]. The question is who chooses to engage in heterophilic relationships and why, given that avoiding relationships with members of the opposite sex is a sanctioned norm in pre-school [31] and primary school years [32], and that, according to social exchange theory [33], children should tend to maximize gains and minimize costs (avoid sanctions).

### 1.2. Peer Network and Cross-Sex Relationships

The results of the few studies on the conditions for establishing closer acquaintances and friendships with opposite-sex peers in early adolescence have revealed the role of the position occupied by a child in the class hierarchy. Our literature review identified only five studies tackling this problem [34,35,36,37,38], while only one [34] involved students in the age of interest to us (11–12 years old). Based on a sociometric study and self-reporting questionnaires, researchers concluded that both among boys and girls, a high percentage of cross-sex relationships occurred in two opposite groups as far as their positions in the peer network were concerned, i.e., among children with very high and very low levels of acceptance.

This in turn remains consistent with researchers’ findings that children who have the highest position in the peer hierarchy can initiate a change of existing norms [39]. For these children, establishing relationships with other-sex peers can lead to a further increase in popularity in the classroom [40]. In fact, longitudinal studies demonstrate that popularity within same-sex peers leads to increased cross-sex popularity, which in turn increases within same-sex popularity [41]. Peer popularity, in turn, is associated with lower levels of perceived social isolation [42]. Consistent with these findings, it can be expected that for adolescents with a high position in the peer network, establishing relationships with other-sex peers can lead to a further increase in their position in the classroom, and—as consequence—to an improved assessment of their peer relations.

The compensatory mechanism is, in turn, particularly important among children who occupy a peripheral position, as contacts with the other sex act as a buffer against the negative effects of rejection experienced in same-sex group [37]. For children who are already rejected, the possibility of establishing contact with others as a back-up system [34], including even other-sex children, helps to meet their need for intimacy, protecting them from the negative consequences of isolation.

As a result, despite the small number of studies directly addressing this issue, it is reasonable to expect that the process of establishing relationships with members of the other sex should lead to a (deferred) improvement in general well-being, including an improvement in ratings of the quality of peer relationships. There are also other reasons for assuming that establishing relationships with other-sex peers can lead to improved psychosocial well-being. For example, a study from the U.K. found that other-sex relationships, though not as strongly as same-sex relationships, were related to a higher level of general self-esteem [43], while self-esteem was negatively related to perceived quality of peer relationships [44] and other mental health outcomes [45].

The positive relationship between cross-sex relationships and well-being is not, however, claimed in all studies. Recent research among Chinese female adolescents shows that cross-sex relationships are negatively associated with mental health outcomes in the overall sample [46]. These differences in research results may be due, on the one hand, to cultural factors (masculinity and paternalism in Chinese culture), and on the other due to differences in the age of the adolescents investigated. In Zhu et al.’s [46] sample, they were at a relatively younger age (around 13 years) than in Liem and Martin’s research [43] (2.5 years older). The observed inconsistency may then result from adolescents being in different developmental phases.

It is therefore possible that a different mechanism of regulating cross-sex relationships may occur in the later period of early adolescence than in the pre-pubertal phase. In the light of the previous research, it is highly probable that, in the pre-pubertal phase, other-sex likeability is a non-normative behavior/feeling, while from the pubertal phase on, it becomes normative as an important developmental task of adolescents with significant implications for their psychosocial well-being, which is analogous to romantic relationships in the stages of development to follow (Davila, 2008). According to Brown [47], at a pubertal stage, in seeking acceptance in a peer group, adolescents must display (or feign) interest in cross-sex relationships. Consequently, the increased interest in the opposite sex is accompanied by gradual changes in the composition of peer groups, and sex homophily is replaced by mixed groups [48]. In the context of establishing cross-sex relationships, these findings lead to particular attention being paid to the pre- and pubertal periods, i.e., from 8 to 13 year old girls and 9 to 14 year old boys, based predominantly on European data [49,50].

### 1.3. The Sense of Peer Integration and Cross-Sex Relationships

The growing importance of peer relationships during early adolescence means that the failure to establish satisfying relationships becomes more harmful and results in a lower sense of integration [51], lower self-esteem [52], and level of satisfaction with one’s life [53] than in the earlier stages of development. Studies demonstrate that the intensity of the sense of integration gets lower between the first grade of elementary school and mid-adolescence [54,55,56]. Current estimates are that in the period of early adolescence, 7% to 12% of teenagers experience feelings of isolation [55], which can be reduced to a large extent by being liked by peers [57].

Although the sense of peer integration is associated with an objective position in the peer network, it is certainly not identical with it. As stated by Cacioppo et al. [58], the two aspects of relationships—objective and subjective—are neither theoretically nor empirically equivalent. Individuals with a negative perception of their own social relationships are not necessarily socially isolated in an objective sense [59]. Research indicates the existence of an association between the two (subjective and objective) dimensions of social relationships [60], but the strength of the correlation between them is moderate at most, not exceeding the value of 0.4 [61].

The sense of peer integration is the final step in a process [62] in which distress is caused by a perceived discrepancy between the person’s actual social relationships and the standards/expectations of optimal levels of social contact [63]. Thus, consistently with causal and/or transactional models [51], subjective, unacceptable feelings of the poor quality of peer relationships [64] mediate associations between (objective) social isolation and potential negative outcome [65]. As a result, the sense of being isolated contributes more to the appearance of negative emotional states than objective social isolation [66,67]. In this context, studies that focus on the subjective dimension of classroom peer relationships are of particular relevance.

Interestingly, to the best of our knowledge, to date, there have been no empirical studies that would determine if—and if so, in what way—the process of including the other sex in originally homophilic social circles is linked to the dynamics of peer integration. A major limitation of studies linking other-sex relationships with perceived quality of peer relations is their cross-sectional character. Since cross-sectional data represent only one moment in time, they do not meet the “causality” assumption that cause preceding effect. Given this, to make arguments about causality, cross-sectional analysis relies on theoretical inferences [68]. In other words, just because at a given point in time, having other-sex relationships is associated with good ratings of the quality of peer relationships, it does not mean that establishing them (including with people of other sex in one’s peer networks) triggers a process of improving quality ratings. At this stage, our knowledge in this area is based on indirect conclusions [24] rather than reported facts [69]. So, in spite of the theoretically supported empirical reasoning, there is no empirical evidence. The present study is unique in that it utilizes a longitudinal framework to examine these associations over time.

### 1.4. Research Problems and Hypotheses

The main purpose of the study was to show the dynamics of the inclusion of opposite-sex peers in networks during early adolescence, i.e., between grades 5 and 6 (in Poland, children aged 11 to 13) and to examine the relationships between the process of including the opposite sex, and changes in the sense of integration with classroom peers.

In the first step of the analyses, we examined whether the presence of strong gender homophily was confirmed in the sample of early adolescents studied—that is, whether at the beginning of grade 5, the clear majority of both female and male adolescents would nominate same-sex peers as “liked”. This was not treated as the test of our hypotheses, but as confirmation of well-known facts and verification of the representativeness of the study sample. The first hypothesis to be verified through the study was H1: Between the beginning of grade 5 and the end of grade 6 of primary school, the frequency of “liking” across the sexes will increase.

In light of well-documented developmental trends, such an effect was to be expected, but for this rather short two-year developmental period, it was uncertain.

The second hypothesis posed in this study was H2: The appearance of the opposite sex in the peer network between grades 5 and 6 will improve the sense of peer integration in adolescents, in comparison to those who only maintain homophilic sympathies.

This was our original assumption as, to the best of our knowledge, no previous research had explicitly tested such a hypothesis. However, such a relationship might have been expected since breaking the homophilic norm in inter-sex relations during early adolescence could be beneficial primarily for students occupying extreme high and low positions in the group structure [34]. Therefore, supposedly in the case of students with high position, opening up in peer relations to the opposite sex might consolidate their position in the group hierarchy and thus enhance the sense of peer integration. Moreover, pupils occupying a low position were expected to benefit from establishing relationships with members of the other sex, for which they might be a back-up system [34].

Moreover, the very fact of being open to the opposite sex, regardless of one’s position in the group, might improve the sense of integration. Early-adolescence heterophilic relations are governed by the principle of “mutuality”, which is universal for all peer interactions [70]. The more people of the opposite sex you like, the more you are liked by them. On the other hand, it is important to bear in mind research findings which show that the sense of isolation is reduced to a high extent by being liked by same-sex peers [57], and that forming relationships with members of the other sex may be regarded as a violation of prevailing norms—particularly among boys [25,27]. The uncertainty of the hypothesis makes it all the more worth testing. In the planned analyses (cf. The Plan of Analyses), we have included a number of measures of group position in order to check whether their inclusion will bear out a possible positive relationship between openness to the opposite sex and the sense of peer integration. Furthermore, due to the already mentioned specific character of girls’ and boys’ development, these relations may look different depending on the sex of the respondent. In order to identify any possible differences, all analyses have been carried out with reference to the division into boys and girls.

## 2. Materials and Methods

### 2.1. Participants

Data were drawn from the longitudinal study ‘School Effectiveness Research—SER’ [71,72], based on a representative sample of adolescents from Polish primary schools. The study used a stratified two-stage cluster sampling procedure. The strata were determined by type of urbanization and the number of class units in a school. Within the strata, schools were sampled with a probability proportional to size (number of students). The analyses considered 5748 students in total—of whom 49.9% were girls—from 288 classes of 169 schools. The mean number of pupils in a class was 20.1; the mean age of our respondents in grade 5 (T1) was 11.3, and in grade 6 (T2) it was 12.8. The interval between T1 and T2 was 1.5 years.

Students from grade 5 in one school year (T1) and the same students in grade 6 the next school year (T2) participated in the survey. The first wave was conducted in the first semester of the 5th grade (academic year 2012/2013), the second at the end of the 6th grade (academic year 2013/2014). The data were collected in the presence of a trained interviewer/instructor using a paper and pencil questionnaire during school lessons. The written consent of students and parents was obtained before the survey.

### 2.2. Measures

#### 2.2.1. Measuring Heterophilic Preferences

The tendency towards heterophily was determined on the basis of the answers given by the adolescents to the question: “Who do you like in your class?” Respondents could name an unlimited number of peers. Same-sex as well as cross-sex nominations were allowed. The question was asked in both waves of the study (at the beginning of grade 5 and at the end of grade 6) and, based on the answers, each respondent was classified into one of the four groups: (1) those who did not nominate any peers of the opposite sex at the beginning of grade 5 or at the end of grade 6 (0–0; Stability without OS); (2) those who did not nominate peers of the opposite sex in the first wave of the study but declared such “likes” later in the second wave (0–1; Profit); (3) those who liked children of the other sex at the beginning of grade 5, but did not declare the same type of “like” in grade 6 (1–0; Loss); (4) and those who showed heterophilic tendencies both in grade 5 and grade 6 (1–1; Stability with OS). The categorical variable created in this way reflects the changes occurring in peer relationships, taking into account students’ sex.

#### 2.2.2. Other Indicators of Peer Relationships

In the predictive analyses, three centrality measures were also used to provide additional information about the structure of each student’s peer relationships, separately in grades 5 and 6. The indicator of how much a student is liked by the same-sex (same-sex in-degree, hereafter referred to as S-S Ind) and the other sex (other-sex in-degree; O-S Ind) was included. Additionally, the models used a variable that indicated how many same-sex peers were liked by a respondent (same-sex out-degree; S-S Out).

In-degree centrality [73] is given as a ratio of the sum of nominations received by students to the number of all possible nominations that could be received. By analogy, out-degree centrality is a ratio of the sum of nominations given by student to the number of all peers in the classroom minus 1. Let us assume that we have a network like the one presented in Panel A of Figure 1. Student A nominates only student B. Students C nominate students A, B, and D. Student E nominates student A, but nobody nominates him/her. Panel B represents an adjacency matrix that contains the same information as the graph in panel A. From the lines, we can read who the group members have chosen and from the columns, who was chosen by whom. For example, student B selected A (1 in column A). He himself was chosen by A (1 in row A of column B) and by C (1 in row C column B). Panel C shows the number of received and given nominations (column and row, respectively) and the values of in-degree and out-degree measures created by dividing the appropriate number of nominations by 4, that is, the number of group members minus 1 (see Figure 1).

As a result, in-degree centrality means the degree of relations that person X receives from others, and it is based on the number of incoming links. Students with high in-degree have more—compared to class peers—choices received from others and are more likeable than others. On the other hand, out-degree centrality, based on the number of outgoing links, means the degree of relations that person X sends toward others. In other words, the student with high out-degree is more active in choosing others, and a lot—compared to other classmates—of other students are liked by him or her. In-degree and out-degree values range from 0 to 1. The values of sociometric indices (O-S Ind; O-S Out; S-S Ind; S-S Out) were calculated in the Igraph package [74].

#### 2.2.3. Measuring the (Subjective) Sense of Peer Integration (PIQ-SI)

The Perceptions of Inclusion Questionnaire—Social Inclusion (PIQ-SI) is a component of a larger self-assessment tool used for measuring student integration in school FDI 4–6 [75]. The Polish adaptation developed by G. Szumski [76] was used to measure the sense of peer integration. The abbreviated version of the PIQ-SI consists of 4 items (examples: “I have very good relationships with my classmates”) to which interviewees respond using a 4–point scale ranging from 1 (not true) to 4 (completely true). A higher score signified a higher level of satisfaction with peer relationships. The shortened version of the PIQ-SI demonstrates reliability and validity indices equivalent to the original longer (15 items) version [77,78]. The PIQ-SI scaling method is presented in The Plan of Analyses section.

### 2.3. The Plan of Analyses

The results of the analyses are presented in two groups; the preliminary analyses focus on methodological issues, while the main analyses center on hypothesis verification.

#### 2.3.1. Verification of the One-Dimensionality and Reliability of the PIQ-SI Scale

Considering that the short version of the PIQ-SI questionnaire had never been verified in Poland due to its psychometric properties, the statistical analyses included the verification of the one-factor structure of the scale as assumed by the authors. For this purpose, we used confirmatory factor analysis. Three measures were used to assess the fit of the model to the data: (1) root mean square of approximation, RMSEA; (2) the Tucker-Lewis index, TLI; and (3) the comparative fit index, CFI. According to generally accepted rules [79], the model indicating an adequate fit to the data should have RMSEA values equal to or less than 0.06, as well as CFI and TLI values greater than 0.90. At the same time, lower values of RMSEA and higher values of CFI and TLI coefficients testify to the better fit of the model to the data and more adequate reflection of the “actual” factor structure.

Cronbach’s α [80] and ω coefficients were used to test the reliability of the scale [81]. The possible values of the two measures range from 0 to 1. Unlike the α coefficient, ω does not assume tau-equivalence, i.e., equality of factor loadings of the scale positions, and it is recommended for testing the reliability of latent variables. The measure is considered reliable when ω > 0.7 [82].

#### 2.3.2. Testing Measurement Invariance (in View of Time and Sex)

The next step in the analysis involved the verification of measurement invariance of the PIQ-SI scale. Measurement invariance is a critical assumption in any longitudinal or intergroup comparison [83]. There is a general consensus among researchers that, for example, the comparisons of differences between the means make sense if at least partial scalar invariance is obtained [84].

The procedure of sequential estimation of a series of hierarchically nested models with an increasing number of constraints was applied to test measurement invariance. The first tested model was a configuration model (M0), that is to say, a model in which no tool invariance is assumed (all model parameters in each period were tested as potentially independent from each other). In the second step, the metric model (M1) was estimated, with a constraint on the magnitude of factor loadings (so-called weak invariance). In the third step, a scalar model (M2) was estimated, in which in addition to the magnitude of factor loadings, there was also equality constraint imposed on the corresponding thresholds of the theorems. The thresholds define transitions from one category of an ordinal variable to another on a normal distribution and play a similar role as means for quantitative variables.

In order to check whether the imposed constraints significantly impair the model’s fit to the data in relation to the unrestricted model, a change to two measures—*CFI* and *RMSEA*—was included, as proposed by Meade et al. [85]. We accepted the rule that the hypothesis of measurement invariance would be rejected when Δ*CFI* exceeded 0.002 or Δ*RMSEA* was greater than 0.007. Obtaining at least partial scalar invariance was the starting point for making comparisons between the latent means describing the level of the variable of interest to us. The plan assumed the verification of measurement invariance, both longitudinal (due to time) and intergroup (due to gender).

#### 2.3.3. Latent Difference Score Model

The latent difference score model (LDS) proposed by J.J. McArdle was used in order to estimate the changes in PIQ-SI intensity [86,87]. The change between the score obtained at Time t2 and Time t1 (LDS_2-1_ = Y_2_−Y_1_) is estimated [88] using the regression equation:Y_2_ = 0 + 1·LDS_2−1_ + 1;(1)

A graphic illustration of such a model is shown in Figure 2. In the LDS model, two regression coefficients (Y_1_ → Y_2_ and LDS_2−1_ → Y_2_) have values set to 1 (one), and Y_2_ is set to 0 (zero), similarly to the mean and the variance of residual errors Y_2_. Determining the path Y_1_ → Y_2_ assumes that some of the score of t_2_ is equal to that of t_1_. As a consequence, LDS_21_ (residual variable) can be interpreted directly as a difference (Δ) in the obtained score between waves. More specifically, Δy is a part of the score Y_2_, which is not identical to Y_1_ and can be estimated as a latent feature having the mean and variance.

From the point of view of the analyses planned, it was important that the LDS model allowed not only for the description of the changes occurring in the intensity of a phenomenon, but also provided the opportunity to examine the influence of independent variables on the initial and final state, and the change in the phenomenon of interest to us [89]. It was also important that the determining factors might be temporally unchanging, (“time-invariant covariate”—TIC), e.g., the sex of pupils during the study, or take changing values (“time-varying covariate”—TVC) in different phases of the study, e.g., a change in heterophilic relationships between grades 5 and 6. Taking into account the TIC variables results in the change to be calculated as a net effect, estimated after excluding the influence of effects exerted by the independent variable. This in turn allowed us to check what effect the change had on the independent variable TVC when the impact of TIC variables was controlled for.

Three models were estimated as planned. The first did not include any predictors (see Figure 3, panel 0), allowing for the estimation of the character of changes in the sense of peer integration occurring between grades 5 and 6. The next two differed, with a set of independent variables included. In the next model (see Figure 3, panel 1), we set out to verify whether the changes in heterophilic sympathies occurring between grades 5 and 6 affected the changes in the sense of peer integration. In the next model (see Figure 3, panel 2), we planned to check whether the anticipated impact of changes in relationships with the opposite sex would persist after the introduction of three additional indices of peer relationships (S-S Ind; O-S Ind; S-S Out).

It should be noted that the variable Other-Sex Out-degree (O-S Out) was not included in the regression model. This was because the information included in the O-S Out variable (student indicates someone of the other gender or does not indicate) was partly used to construct a categorical variable describing the cross-sex relationships occurring between 5th and 6th grade (Stability without O-S, Stability with O-S, Profit O-S, and Loss O-S), a key predictor of changes in the sense of quality of peer relationships (cf. Figure 3). Since both variables share overlapping information, including OS-Out in the regression model could result in biased estimation of key independent effects predictor variables on the outcome variable [90]. All analyses were carried out in multigroup models, simultaneously for girls and boys.

#### 2.3.4. Software and Estimation Methods

The Mplus package (version 8.0) was used for the analysis of factor structure and LDS models. Because the PIQ-SI questionnaire is measured using a Likert-type scale, all models were estimated based on the polychoric correlation matrix and weighted least squares mean-and-variance-adjusted (WLSMV) [91], as recommended for categorical (ordinal) data [92]. Since the analyzed data were hierarchical—as children were nested in classes—the Complex sample option in Mplus was used to avoid bias to standard errors and test statistics.

The level of missing data on the PIQ-SI was low in both waves. In Wave 1, there were 66 missing values distributed over 4 items (from 12 missing values in Item 2 to 24 missing values in Items 3). In Wave 2, there were 63 missing values over items (from 8 missing values in Item 2 to 30 missing values in Item 4). Little’s [93] missing completely at random (MCAR) test was conducted to test whether data were MCAR. It supports the hypothesis that data were MCAR, both for the first wave, *χ*^2^ (16) = 25.122, *p* = 0.07, and the second wave, *χ*^2^ (15) = 17.8, *p* = 0.28, and across all data, *χ*^2^ (103) = 126.28, *p* = 0.06. In consequence, the missing data were handled with full information maximum likelihood (FIML) estimation [94].

Completion rates in the classroom for the peer nomination instrument (sociometry) were (minimum) 60.71 in the first wave (*M* = 87.5) and 60.10 in the second wave (*M* = 86.81). A participation rate of at least 60%, in conjunction with the procedure of unlimited nominations, provides a stable estimation of status in the peer group [95].

#### 2.3.5. Categorical Data Analysis

Analyzing the relationship between categorical variables, e.g., differences between boys and girls in the intensity of heterophilic preferences (in grades 5 and 6), we used a chi-square independence test [96]. For contingency tables larger than 2-by-2, adjusted standardized residuals (ASR) were also used [97]. These coefficients indicate whether its observed frequency is significantly different from the expected frequency. In consequence, ASR can be used to describe the pattern of association among the table cells and may be treated as a post hoc analysis of the chi-square (omnibus) test. Although the low level of statistical significance of the chi-square test shows that there are some significant differences in the contingency tables, the ASR indicates which cells are different from each other. As ASR has an asymptotic standard normal distribution; an absolute value greater than 1.96 indicates a deviation from independence in the cell. If the ASR is less than −1.96, the observed frequency of the cell is lower than expected. If the ASR is greater than +1.96, the observed frequency is higher than expected [98].

To test the significance of differences intensity of heterophilic preferences between grade 5 and grade 6, we used McNemar’s test [99]. McNemar’s test is a chi-square test that is used to compare two proportions when the data are paired (that is, measured on the same sample). To distinguish between McNemar’s chi-square and regular chi-square independence tests, we used a prefix “*M*” denoting McNemar’s test (e.g., *Mχ*^2^).

## 3. Results

### 3.1. Preliminary Analysis

Table 1 shows the correlations among variables used in the analyses at the two time points.

#### 3.1.1. Factor Structure and Reliability of the PIQ-SI Scale

The initially tested confirmatory factor model assuming the occurrence of one latent variable proved to be poorly fitted to the data (see Table 2), as reflected in the relatively high value of the RMSEA statistics (significantly exceeding the limit of 0.06). The inspection of the modifiable indexes showed that adjusting the model would improve the correlations of items 2 (“I get along very well with my classmates”) and 4 (“I have very good relationships with my classmates”). The introduction of this correction clearly improved the measures of goodness of fit. For example, among all students (i.e., without consideration given to their sex), RMSEA decreased from 0.108 to 0.059. Interestingly, this happened in the case of data from grades 5 and 6 (both in the case of all respondents and taking sex into account). This indicates a stable (longitudinal and inter-group) content redundancy of both statements. The subsequent analyses will use a model assuming the occurrence of correlations between the aforementioned positions.

The tested version of the scale showed a satisfactory level of internal consistency. Cronbach’s alpha in grade 5 was 0.80 (girls 0.80, boys 0.79), and in grade 6, it was 0.81 (girls 0.82, boys 0.80). The ω coefficient in grade 5 was equal to 0.86 (both in boys and girls), and in grade 6, it was 0.88 (girls: 0.89, boys: 0.87).

#### 3.1.2. Measurement Invariance—PIQ-SI Scale

The next step in our analyses focused on verifying the assumption of the PIQ-SI scale measurement invariance. Two groups of models were tested: (1) those assuming only longitudinal invariance (without specifying gender), and (2) those considering invariance simultaneously in view of time and gender (see Table 3). In both cases, the tool demonstrated configural and metric invariance. Obtaining scalar invariance required the freeing of some of the model parameters.

When testing invariance only over time, modification indices showed the necessity of setting the third threshold of item 1 free. In turn, invariance both over time and in view of gender required the freeing of four parameters: the third threshold in grade 5 among boys, the third threshold of item 4 also in a group of boys in grade 5, the third threshold of item 2 among boys in grade 5, and the third threshold of item 1 among girls in grade 5 (the analyses indicating changes in the model fit were carried out sequentially). Further analyses were consequently conducted on the basis of partial scalar measurement invariance models (both in the case of all respondents, as well as respondents divided on the basis of their sex).

Establishing partial scalar invariance made it possible to compare the mean differences in the intensity of the sense of social integration between grades 5 and 6 both among all respondents and among female and male students. The comparison of the means indicated a small decrease (in standard deviation of 0.16) in the sense of peer integration. However, when sex was considered, it turned out that the drop occurred only in the case of girls.

#### 3.1.3. Unconditional Latent Difference Scores Model (LDS) of the PIQ-SI Scale

The estimation of the latent difference model confirmed the conclusions mentioned above (see Table 4). The difference between the sense of integration in grades 5 and 6 was proven to be statistically significant (and negative) among girls, but not boys.

This trend should be kept in mind when interpreting the results of the main analyses.

### 3.2. Main Analysis

Basic information on sex heterophily is shown in Figure 4. As expected, a significant proportion of students did not nominate a peer of the opposite sex either in grade 5 or 6 (see Figure 4—Panel A). This result confirms a well acknowledged regularity and thus also confirms the representativeness of the sample in this respect.

Let us proceed to testing hypothesis 1. An increase in the intensity of heterophilic preferences (HP) was observed (*Mχ*^2^ (1) = 143.59, *p* < 0.01); in grade 5, it amounted to 43.5%, while at the end of grade 6, it was 53.1%. The group of female and male students who declared a liking for at least one person of the other sex was growing. This result confirms H1.

Moreover, the results show that boys are more homophilic (less heterophilic) than girls (for grade 5: *χ^2^* (1) = 35.10, *p* < 0.01; for grade 6: *χ^2^* (1) = 20.58, *p* < 0.01), as indicated in both waves of the study (see Figure 4, Panel B and D). While in grade 5, 60.5% of boys did not nominate any girl among the peers they liked (among girls, the equivalent figure was 52.5%), in grade 6, the corresponding number was 50.1% for boys and 43.8% for girls. In the period of 1.5 years, in both sex groups, there was an approximate 10% drop in adolescents declaring an exclusive liking for same-sex peers. These changes are statistically significant for boys (*Mχ^2^* (1) = 83.62, *p* < 0.01) and girls (*Mχ^2^* (1) = 60.02, *p* < 0.01).

At the same time, the analysis of the direction of change taking place in heterophilic relationships (see Figure 4, Panel C) indicates that boys do not differ significantly from girls as far as two “dynamic” categories (Profit and Loss) are concerned. In both groups, we found that there was a similar percentage of students who (1) in grade 6 declared their liking of a person or persons of the opposite sex, although in grade 5, their likeability preferences were homogenous and included only same-sex peers (Profit; for boys adjusted standardized residuals (*ASR*) were equal 1.8; for girls *ASR* = –1.8)), and (2) in grade 6 did not mention they liked a peer or peers of the opposite sex, although in grade 5, such persons were represented in their likeability network (Loss; *ASR_Boys_* = –0.2; *ASR_Girls_* = 0.2). It is worth emphasizing that the Profit indicator is approximately twice as high as the Loss one among both boys and girls. 

However, clear differences were observed in the case of both “stable” categories (Stability without OS and Stability with OS). Boys more often than girls did not declare liking an other-sex peer, both in grade 5 and in grade 6 (Stability without OS): 38.1% (*ASR_Boys_* = 4.6) against 32.0% (*ASR_Girls_* = –4.6). Girls more often (*ASR_Girls_* = 6.4) than boys (*ASR_Boys_* = –6.4) declared that they liked someone from a different sex group in both waves (Stability with OS: 35.7% vs. 27.5%). The dynamics of changes in heterophilic preferences are therefore similar among boys and girls. Overall, the cross-sex differences consisted of greater homophily among boys.

The second hypothesis (H2) was tested with the use of conditional latent difference score (LDS) models of the PIQ-SI scale. In the first model that considered the predictors for regression analysis, only the variables that reflected the changes in heterophilic preferences among adolescents between the beginning of grade 5 and the end of grade 6 were introduced. The analysis showed that being liked by a person of the opposite sex (Profit) and maintaining this (Stability with OS) were the significant predictors that improved the change in the sense of peer integration. This applied to both girls and boys (see Table 5—Model 1 and Figure 5).

Three additional indices (SS-Ind; OS-Ind; SS-Out) of peer relationships were introduced in the next model (see Table 5—Model 2 and Figure 5) to see if the impact of the changes taking place in heterophilic preferences is maintained with other dimensions of peer relationships controlled for. The results confirmed that when they were, the establishing of peer relationships with adolescents of the opposite sex increased the sense of peer integration, although the magnitude of the regression coefficients dropped from 0.35 to 0.16 among girls and from 0.38 to 0.16 among boys. This result confirms the hypothesis according to which the appearance of the opposite sex in the peer network between grades 5 and 6 will improve the sense of peer integration (H2). This effect is partially mediated by various measures of the student’s peer network position, but a direct effect unmediated by network position, of including peers of the opposite sex in the networks due to a sense of peer integration, is also present.

The model analyzed yields additional interesting information. In grade 6, the satisfaction with peer relationships—both among girls and boys—is associated with the higher number of people of the same [S-S Ind GR6] or other [O-S Ind GR6] sex who liked the respondent. In grade 5, this effect was found only among boys, although among girls, the regression coefficient is also positive. So, after controlling other variables, liking other-sex peers is associated with satisfaction with the peer relationship (especially for boys). Taken together, these findings suggest that not only liking other-sex peers, but also being liked by them, promotes a positive perception of peer relationships. This (indirectly) indicates the existence of a reciprocity mechanism: the more people of the opposite sex you like, the more you are liked by them.

## 4. Discussion

The main goal of the study presented was to examine the dynamics of including other-sex children in peer networks, and the relationship between these changes in the sense of peer integration across 1.5 years in early adolescents, based on a two-wave set of longitudinal data from a large representative sample (n = 5748) of Polish primary school pupils.

The study confirmed that at the threshold of adolescence, interactions and likeability among Polish early adolescents were still taking place primarily within homogenous groups, whereas cross-sex relations were quite rare. The higher level of boys’ homophily revealed in this study is consistent with the results of other research [100]. This is related to a higher, relative to that of girls, self-perception through references to the characteristics typical to their own sex, and the treatment of their same-sex group as better, along with higher peer pressure to observe the homophilic norm [101]. As reported in earlier studies [102], an additional factor contributing to higher homophily among boys may also be the fact that overall, boys have lower social skills necessary to establish relationships with other-sex peers [36,103]. The results of previous studies led to the conclusion that girls form more new relationships than boys [11,13,14,15]. Generally, the findings of our study stay in line with the results of prior research showing that homophilic tendencies appear among boys later than among girls [7], but are stronger and more durable.

Our results also provide a better understanding of another development trend. The process of breaking down sex homophily among early adolescents is likely to be related to the genetically and hormonally conditioned biological maturation linked to sexual development [104] and a growing interest in other-sex relationships, grounding the development of romantic relationships. According to Brown [47], the first phase of romantic relationships is the “initiation phase”, characterized by the establishment of cross-sex relationships, associated with the development of specific social and personal skills.

In fact, research to date shows that having more cross-sex friends increases chances of an adolescent making their romantic relationship debut [105]. However, research also suggests that this debut did not originate from cross-sex friends [106]. Among early adolescents, cross-sex relationships “function as indirect ‘training grounds’ for romantic relationships where adolescents learn how to interact with the cross-sex peers rather than a dating pool in and of itself” [105]. As such, cross-sex sympathies form important contexts in which adolescents probably learn to interact with other-sex peers. Researchers suggest that other-sex relationships allow adolescents to practice and refine those skills that are later used to build and maintain romantic unions [107]. Note that adolescents themselves have reported some unique benefits of other-sex relationships, such as the opportunity to see others’ perspectives and to learn about other-sex relationship expectations [108].

For the development of peer relations theory, it is particularly important to confirm hypothesis 2, according to which the appearance of the opposite sex in the peer network between grades 5 and 6 will improve the sense of peer integration. With the Latent Difference Score (LDS) model with additional network measures taken into account, the force of the impact of relationships with the opposite sex on the sense of peer integration weakens, though it does not cease to be a significant predictor.

Our results indirectly support the idea that establishing other-sex peer relationships can result in positive feedback. The tendency to reciprocate others’ emotions and actions is a universal feature of social life [109]. Moreover, among adolescents’ positive peer relationships (e.g., friendship), this means mutual liking and reciprocity [110]. Thus, it can be expected that liking other-sex peers will be associated with higher levels of being liked by other-sex peers. This mechanism may increase perceived social support [111] which—as demonstrated by numerous studies [112]—improves well-being in children and adolescents, including their evaluation of the quality of peer relationships [113].

Confirmation of this hypothesis suggests a positive role for reducing structural (institutional) barriers to create other-sex relationships as a way to improve perceptions of the quality of peer relationships. This is also in line with the results of previous research showing that when access to other-sex peers is foreclosed, the range of beneficial functions of other-sex peer relationships can be limited [114]. So, limiting peer relationships to same-sex partners minimizes access to the benefits of being accepted by the other sex.

Limiting relationships with other-sex peers can have particularly negative consequences for adolescents who are rejected by same-sex peers: “For these children, positive and supportive other-sex peer relationships may provide an important protective refuge. Thus, for such children, opportunities for interactions and relationships with other-sex peers may have the potential to reduce the negative effects of problems with same-gender peers” [114]. In turn, this can—in light of our research—translate into a better view of the quality of peer relationships, also reinforcing self-esteem [115].

It seems that building cross-sex peer relations can be particularly useful for gay adolescents since it is known that heterosexual girls tend to express less prejudice than heterosexual boys towards gay boys, and thus, cross-sex friendship may provide for gay adolescents an opportunity to establish a larger social support network [116]. Further to this, Bowker and White [117] also draw attention to the psychological benefits of other-sex relations for withdrawn boys, “because their behaviors may be more consistent with gender norms for girls, especially during middle childhood and early adolescence when gender norms tend to become more flexible” [117].

The review of Mehta and Strough [12] concluded that there are several causes that contribute to the reduction in cross-sex relations during adolescence. One of the most important of these is institutional barriers, including the structure of schools and classrooms. School-based interventions to reduce these barriers could include not only educating children about and challenging gender stereotypes, and exploring perceived similarities between genders [118], but also rearranged classroom settings [119,120] and reducing the interpersonal distance between other-sex adolescents [121]. Reducing such barriers can create better environments for the successful development of adolescents, since being liked by peers is a key developmental task for young people and a sign of positive adaptation [122].

These findings must be interpreted in light of several limitations. It should be noted that our research was limited only to school peer networks. It is known that adolescents’ activities with peers are not limited to the school environment [123]. Therefore, it would also be worth taking into account out-of-school peer relations in subsequent studies. Nevertheless, in Poland, 87.2% of the liked colleagues were peers from the same school class, 5.5% were people from the school (although outside the school class), and only 7.3% were people from outside the school. This demonstrates the importance of the educational system as a meaningful context in which adolescents spend most of their time.

The effect of relationships with the opposite sex on a changed sense of peer integration was only tested with two-wave data. Future studies may use latent growth curve (LGC) models [124] based on three or more waves of data in order to examine the trend over a longer period of time and with more sensitive methods.

Our study also did not include contextual variables, such as the sex structure of the class. Prior research suggests that (a) having more opposite-sex classmates led to more cross-sex relationships [125] and (b) that people identified least with a group when their own gender differed from that of other group members—when a person is in the minority based on the gender structure of the group [126]. Previous research also indicates that individuals in more homophobic peer groups engaged in even more homophobic behavior than can be accounted for based solely on their own individual prejudice attitudes [127]. Moreover, research on friendship homophily among immigrants highlights the importance of contextual variables. A higher percentage of immigrants at school is related to a higher degree of co-ethnic friendship homophily [128]. Unfortunately, including contextual data requires the use of multi-level modeling, which limits the possibility of using latent variables (due to small class sizes) and results in reduced measurement reliability.

Furthermore, we did not measure of the sexual identity (self-identification as gay, lesbian, or bisexual) of our respondents. Previous studies have shown that sexual minority adolescents show lower levels of peer acceptance [129], rated their relationships with peers less positively than others [130], and are characterized by a slightly lower level of peer homophily [116]. So, it is possible that those in our research who established other-sex relationships and experienced an improved perception of the quality of peer relationships are members of sexual minority groups. However, this does not affect the conclusion that forming other-sex relationships improves perceptions of the quality of peer relationships.

Finally, the generalizability of the findings is limited because the study was carried out in Poland. The strong attachment in Polish society to traditionally defined sex roles [131] suggests that establishing peer relations with the opposite sex in early adolescence may clash with a stronger homophilic norm. The importance of cultural context as a factor influencing the initiation of inter-sex relationships during adolescence has been researched by Zhou, Li and Wang [46].

In Poland, the academic understanding of the category of “gender” has been contrasted sharply with the political construct of “gender ideology”. While the concept of gender is part of the conceptual framework of social research, the latter was created and successfully mainstreamed into political discourse in Poland in the second decade of the 21st century [132,133]. Recent years have seen the dispute driven by this ideology becoming aggravated and transferred to the school space [134]. Also relevant to this aspect of peer relations in Poland may be the weakness of school-based or informal sex education, which may keep other-sex peer relations taboo [135]. All this may be perceived by adolescents as an encouragement to stay longer in same-sex peer relationships. Nevertheless, since most research on peer relationships is conducted in the U.S. and Western Europe, it is useful to add to the pool of findings a large, representative study from another part of the world.

## 5. Conclusions

It is a well-known fact that in adolescence, the importance of peer relationships increases significantly, and the sense of peer integration is associated with solving developmental tasks and psychological and social well-being. Previous studies confirm that in the stage of early adolescence, in peer interactions, choosing same-sex friends is the dominant trend, although it is beginning to be accompanied by the appearance of cross-sex choices (sex-related heterophily within peer relations), which are a preliminary stage for the development of romantic relationships. In general, the results of the present study converge with previous theory and research indicating that the level of heterophily (choosing peers of the other sex as liked) is higher in grades 5 and 6 among girls than among boys. Our study also confirmed that between the ages of 11 and 13, the intensity of heterophily increases significantly (by approximately 10 percentage points). While in grade 5, only 39.5% of the boys liked at least one person of the other sex, in grade 6, this value was already 49.4%. For girls, on the other hand, the level of heterophily increased from 47.5% to 56.2%.

However, the main objective of the present paper was to determine the relationship between changes in heterophily and changes in perception of the quality of peer relationships. To the best of our knowledge, to date, there have been no empirical studies based on longitudinal data that would determine if, and if so in what way, the process of including the other sex in originally homophilic social circles is linked to the dynamics of the perceived quality of peer relationships. In the cross-sectional perspective, correlational analyses showed that both liking other-sex peers (Other-Sex Out-degree) and being liked by them (Other-Sex In-degree) are associated with better perceptions of peer relationships, as are liking same-sex peers (Same-Sex Out-degree) and being liked by them (Other-Sex-In-degree).

Importantly, our research adds to the previous empirical evidence from a longitudinal study that the process of inclusion (between the 5th and 6th grades) of other-sex peers in initially same-sex peer relationships translates into a better perception of the quality of peer relationships (both among boys and girls). The results additionally indicate that in the case of boys, the loss of relationships with girls has a negative impact on the sense of peer integration. In this way, our findings support the hypothesis [136] that the process of establishing relationships with other-sex peers serves an important developmental function by expanding the pool of attachment figures available to meet the social and emotional needs, and that this process may improve not only the perceived quality of peer relationships, but also broader psychosocial well-being, especially for early adolescents with a deficit in positive same-sex peer relationships.

## Figures and Tables

**Figure 1 ijerph-19-14971-f001:**
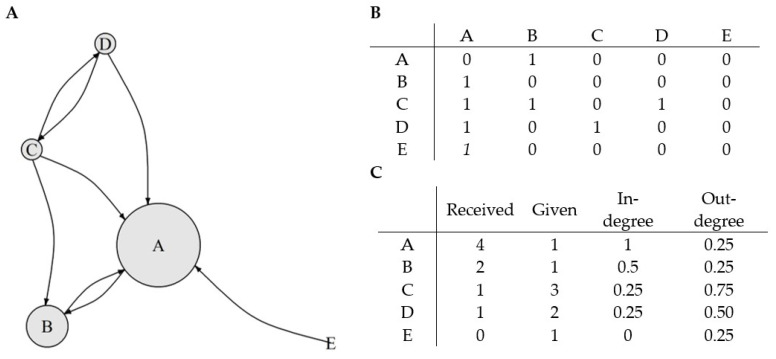
Example of calculation of in-degree and out-degree measures. (**A**) Assumed network; (**B**) adjacency matrix of the network; (**C**) the number of received and given nominations and the values of in-degree and out-degree measures for students in the network.

**Figure 2 ijerph-19-14971-f002:**
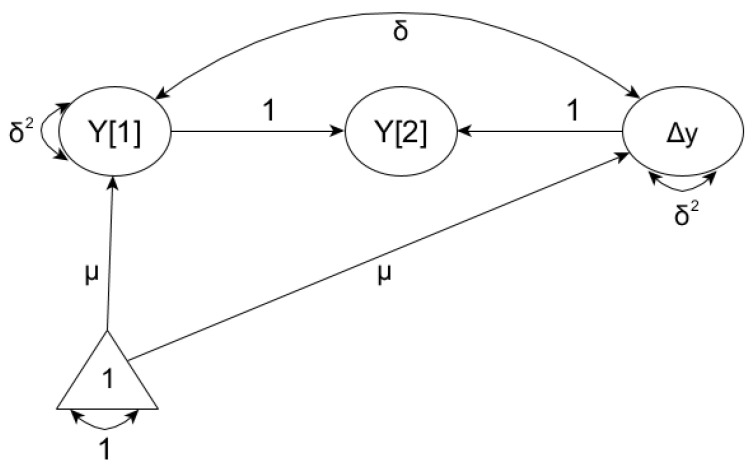
Latent difference score scheme (LDS) for the latent variable measured at two different times. Note: The change between Y_1_ and Y_2_ is estimated as a constant Δy (LDS); the lines with two arrows indicate correlations; the lines with one arrow are regression coefficients; the autoregressive path Y_1_ → Y_2_ and regression coefficient Δy → Y_2_ have values set to 1 (one); the error associated with variable Y_2_ and its constant are set to 0 (zero). The Δy variable is directly unobservable and estimated based on the transformation of the formula: Y [2] = Y [1] + Δy. The parameter δ2 means variances, δ is the correlation, and μ means. The triangle represents the means’ structure.

**Figure 3 ijerph-19-14971-f003:**
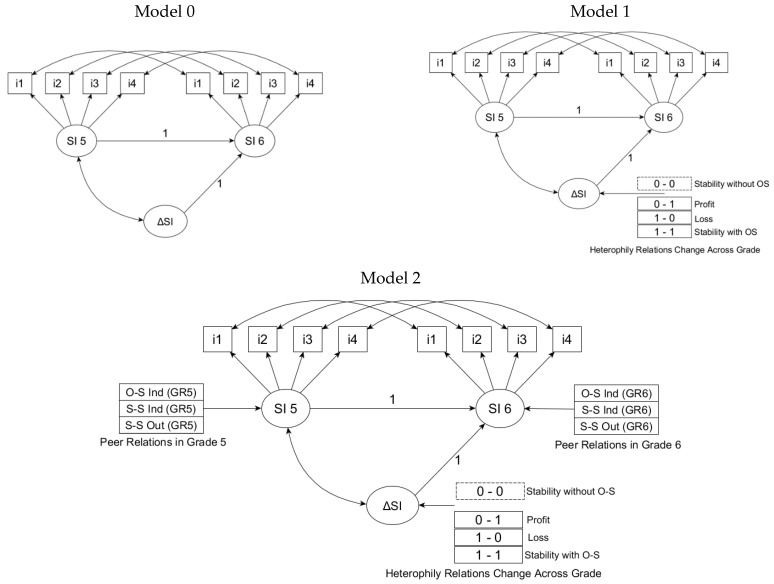
The schemes of the tested latent difference score (LDS) models for the latent variable PIQ-SI measured at two points in time (Grade 5 and Grade 6). Model 0 = unconditional LDS model. Model 1 = conditional LDS model (with predictors that reflected the changes in heterophilic preferences among adolescents between the beginning of grade 5 and the end of grade 6. Model 2 = conditional LDS model (with three additional indices of peer relationships. S-S = same sex; O-S = other sex; Out = out-degree; Ind = in-degree; GR = grade. SI 5 = Sense of peer integration in grade 5; SI 6 = Sense of peer integration in grade 6; ΔSI = Difference in sense of peer integration between grade 5 and grade 6 (latent difference score). i = items of the PIQ-SI scale.

**Figure 4 ijerph-19-14971-f004:**
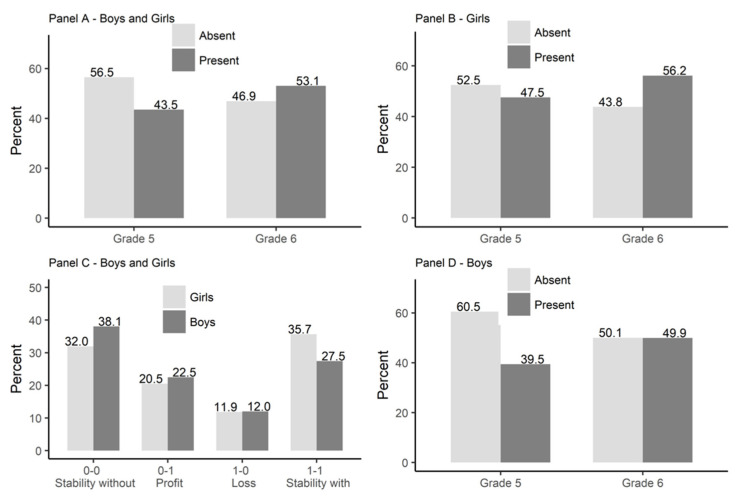
The charts illustrating the distribution of sex heterophily in both waves of the study. Note: Present = declared liking of a peer of the opposite sex; Absent = declared liking only people of their own sex; Stability without = did not nominate any peers of the opposite sex at the beginning of grade 5 and at the end of grade 6); Profit = did not nominate peers of the opposite sex in the first wave of the study but declared such “likes” later on in the second wave; Loss = liked somebody of the other sex at the beginning of grade 5, but did not declare the same type of “like” in grade 6; Stability with = showing heterophilic tendencies both in grade 5 and grade 6.

**Figure 5 ijerph-19-14971-f005:**
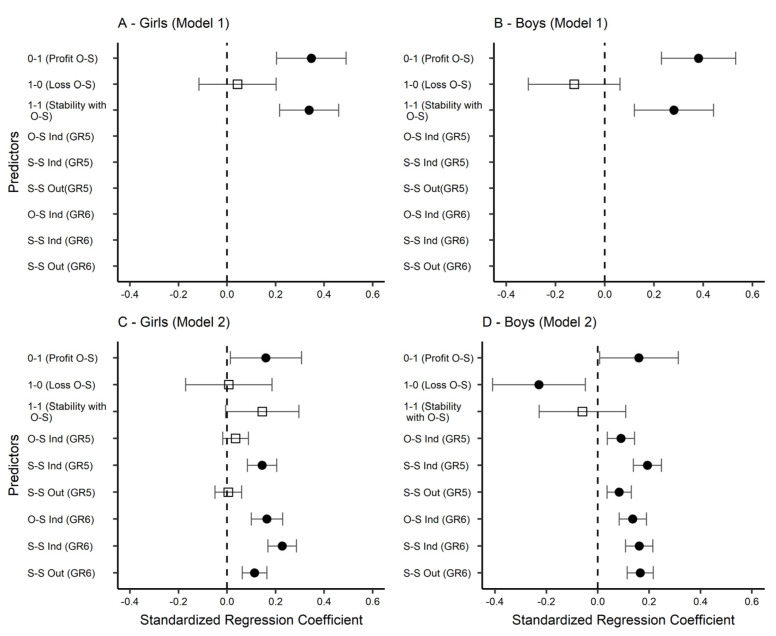
Standardized regression parameters from conditional latent difference score model (LDS) for boys and girls. Note: Black circle—coefficient significant at 0.05; white square—coefficient insignificant (*p* > 0.05); tails = standard errors. S-S = same sex; O-S = other sex; Out = out-degree; Ind = in-degree; GR = grade. Stability without O-S is not presented in the model because it serves as the reference category.

**Table 1 ijerph-19-14971-t001:** Means, standard deviations, and correlations.

Variable	*M*	*SD*	1	2	3	4	5	6	7	8	9	10
*M*			13.70	13.53	0.22	0.22	0.07	0.06	0.21	0.21	0.09	0.08
*SD*			2.51	2.46	0.13	0.12	0.10	0.10	0.13	0.12	0.11	0.11
PIQ-SI (GR5)	13.44	2.50		0.50 **	0.34 **	0.23 **	0.21 **	0.12 **	0.29 **	19 **	0.19 **	0.10 **
2. PIQ-SI (GR6)	13.24	2.58	0.49 **		0.28 **	0.16 **	0.14 **	0.05 *	0.32 **	0.26 **	0.21 **	0.12 **
3. S-S Ind (GR5)	0.23	0.13	0.29 **	0.19 **		0.41 **	0.29 **	0.11 **	0.62 **	0.30 **	0.29 **	0.06 **
4. S-S Out (GR5)	0.23	0.12	0.13 **	0.09 **	0.37 **		0.02	0.24 **	0.30 **	0.40 **	0.07 **	0.09 **
5. O-S Ind (GR5)	0.06	0.08	0.15 **	0.10 **	0.20 **	–0.01		0.43 **	0.21 **	0.03	0.56 **	27 **
6. O-S Out (GR5)	0.07	0.11	0.12 **	0.06 **	0.08 **	0.15 **	0.43 **		0.08 **	0.09 **	0.30 **	0.38 **
7. S-S Ind (GR6)	0.23	0.13	0.22 **	0.29 **	0.60 **	0.28 **	0.14 **	0.04 *		0.42 **	0.34 **	0.09 **
8. S-S Out (GR6)	0.23	0.12	0.14 **	0.19 **	0.28 **	0.44 **	−0.03	0.02	0.42 **		0.06 **	0.22 **
9. O-S Ind (GR6)	0.08	0.10	0.14 **	0.20 **	0.22 **	–0.03	0.53 **	0.28 **	0.26 **	–0.01		0.40 **
10. O-S Out (GR6)	0.09	0.11	0.13 **	0.16 **	0.10 **	0.06 **	0.33 **	0.43 **	0.10 **	0.17 **	0.47 **	

Note: *M* and *SD* are used to represent mean and standard deviation, respectively. * indicates *p* < 0.05. ** indicates *p* < 0.01. PIQ-SI = Social Inclusion subscale from Perceptions of Inclusion Questionnaire. S-S = same sex; O-S = other sex; Out = out-degree; Ind = in-degree; GR = grade. Above the diagonal are data for boys, below for girls. ** Statistical significance level (*p* < 0.01). * Statistical significance level (*p* < 0.05).

**Table 2 ijerph-19-14971-t002:** Measures of fit to the data of the one-factor confirmatory model PIQ-SI.

Model	Par	χ^2^ (*df*)	*RMSEA*	*CFI*	*TLI*
Grade 5	16	131.29 (2) **	0.108	0.990	0.969
Grade 5 ^Mod1^	17	20.22 (1) **	0.059	0.998	0.991
Grade 5—Boys	16	62.63 (2) **	0.104	0.990	0.970
Grade 5—Boys ^Mod1^	17	9.13 (1) **	0.054	0.999	0.992
Grade 5—Girls	16	63.78 (2) **	0.106	0.990	0.971
Grade 5—Girls ^Mod1^	17	10.42 (1) **	0.058	0.999	0.991
Grade 6	16	138.66 (2) **	0.112	0.991	0.973
Grade 6 ^Mod1^	17	15.30 (1) **	0.051	0.999	0.994
Grade 6—Boys	16	54.40 (2) **	0.098	0.993	0.979
Grade 6—Boys ^Mod1^	17	4.71 (1) *	0.037	1.000	0.997
Grade 6—Girls	16	104.08 (2) **	0.137	0.988	0.965
Grade 6—Girls ^Mod1^	17	11.26 (1) **	0.061	0.999	0.993

Note: Par = number of parameters; χ^2^ = chi-square statistics; *df* = degrees of freedom; *RMSEA* = root mean square error of approximation; *CFI* = comparative fit index; *TLI* = Tucker–Lewis index; Mod = modification. ** Statistical significance level (*p* < 0.01). * Statistical significance level (*p* < 0.05).

**Table 3 ijerph-19-14971-t003:** Measures of goodness of fit of the models testing measurement invariance of PIQ-SI across time and in view of gender (simultaneously).

	Model	Par	χ^2^ (*df*)	*RMSEA*	*CFI*	*TLI*	Δ*RMSEA*	Δ*CFI*
Grade	Configural ^a^	39	65.51 (13) **	0.027	0.998	0.996		
	Metric ^b^	36	79.52 (16) **	0.026	0.998	0.996	–0.001	0.000
	Scalar ^c^	29	159.00 (23) **	0.032	0.995	0.994	0.006	–0.003
	Scalar ^c Mod1^	30	126.33 (22) **	0.029	0.996	0.995	0.003	–0.002
Grade and Gender	Configural ^a^	78	72.97 (26) **	0.025	0.998	0.996	–	–
	Metric ^b^	68	111.35 (36) **	0.027	0.997	0.996	0.002	–0.001
	Scalar ^c^	47	338.01 (57) **	0.041	0.989	0.990	0.014	–0.008
	Scalar ^c Mod1^	48	258.74 (56) **	0.035	0.992	0.992	0.008	–0.005
	Scalar ^c Mod2^	49	235.29 (55) **	0.034	0.993	0.993	0.007	–0.004
	Scalar ^c Mod3^	50	204.92 (54) **	0.031	0.994	0.994	0.004	–0.003
	Scalar ^c Mod4^	51	197.81 (53) **	0.031	0.995	0.994	0.004	–0.002

Note: Par = number of parameters; χ^2^ = chi-square statistics; *df* = degrees of freedom; *RMSEA* = root mean square error of approximation; *CFI* = comparative fit index; *TLI* = Tucker–Lewis index. ^a^ Factor loadings and thresholds estimated without constraints. ^b^ Constraints imposed on factor loadings, but thresholds estimated without constraints. ^c^ Constraints imposed both on loadings and thresholds. Mod = modification. ** Statistical significance level (*p* < 0.01).

**Table 4 ijerph-19-14971-t004:** Unconditional latent difference score (LDS) models of the PIQ-SI.

	Boys and Girls ^a^		Girls ^b^		Boys ^b^	
	*Est*	(*SE*)	*Est*	(*SE*)	*Est*	(*SE*)
Mean						
PIQ-SI_GR5_	0 ^F^	0 ^F^	0 ^F^	0 ^F^	0 ^F^	0 ^F^
LDS (Δ PIQ-SI_GR6-GR5_)	–0.157 **	0.031	−0.161 **	0.040	0.089	0.054
Intercept						
PIQ-SI_GR5_	0 ^F^	0 ^F^	0 ^F^	0 ^F^	0 ^F^	0 ^F^
Variance						
PIQ-SI_GR5_	2.001 **	0.128	2.008 **	0.162	1.875 **	0.172
Δ PIQ-SI_GR6-GR5_	1.403 **	0.098	1.588 **	0.141	1.262 **	0.126
Residual Variance						
PIQ-SI_GR6_	0 ^F^	0 ^F^	0 ^F^	0 ^F^	0 ^F^	0 ^F^
Correlation (Δ PIQ-SI_GR6-GR5_~PIQ-SI_GR5_)	–0.467 **	0.021	–0.457 **	0.028	–0.380 **	0.039
Regression PIQ-SI_GR6_ ← PIQ-SI_GR5_	1	1	1	1	1	1

Note: *Est* = estimated parameter; ^a^ = parameters estimated based on the model invariant across time; ^b^ = parameters estimated based on the model invariant across time and in view of sex; PIQ-SI = Social Inclusion subscale from Perceptions of Inclusion Questionnaire; GR = grade. ^F^ = fixed parameter. ** Statistical significance level (*p* < 0.01).

**Table 5 ijerph-19-14971-t005:** Conditional latent difference score (LDS) models of the PIQ-SI scale.

	Model 1				Model 2			
	Girls ^b^		Boys ^b^		Girls ^b^		Boys ^b^	
	Est	(SE)	Est	(SE)	Est	(SE)	Est	(SE)
Mean/Intercept								
ΔPIQ-SI_GR6-GR5_	–0.14 *	0.06	0.17 *	0.08	0.07	0.10	0.05	0.13
PIQ-SI_GR5_	0 ^F^	0 ^F^	0 ^F^	0 ^F^	0 ^F^	0 ^F^	0 ^F^	0 ^F^
PIQ-SI_GR6_	0 ^F^	0 ^F^	0 ^F^	0 ^F^	0 ^F^	0 ^F^	0 ^F^	0 ^F^
Variance/Residual Variance								
ΔPIQ-SI_GR6-GR5_	1.60 **	0.15	1.19 **	0.13	1.66 **	0.19	1.58 **	0.26
PIQ-SI_GR5_	1.90 **	0.16	1.70 **	0.20	1.70 **	0.17	1.96 **	0.40
PIQ-SI_GR6_	0 ^F^	0 ^F^	0 ^F^	0 ^F^	0 ^F^	0 ^F^	0 ^F^	0 ^F^
Correlation								
ΔPIQ-SI_GR6-GR5_~SI_GR5_	–0.45 **	0.03	–0.34 **	0.05	–0.38 **	0.04	–0.44 **	0.07
Regression								
SI_GR6_ ← SI_GR5_	1 ^F^	1 ^F^	1 ^F^	1 ^F^	1 ^F^	1 ^F^	1 ^F^	1 ^F^
ΔPIQ-SI_GR6-GR5_ ← Profit	0.35 **	0.07	0.38 **	0.08	0.16 *	0.08	0.16 *	0.08
ΔPIQ-SI_GR6-GR5_ ← Loss	0.04	0.08	–0.12	0.10	0.01	0.09	–0.23 *	0.09
ΔPIQ-SI_GR6-GR5_ ← Stab with O-S	0.34 **	0.06	0.28 **	0.08	0.15 *	0.08	–0.07	0.09
PIQ-SI_GR5_ ← S-S Out (GR5)					0.01	0.03	0.08 **	0.02
PIQ-SI_GR5_ ← O-S Ind (GR5)					0.04	0.03	0.09 **	0.03
PIQ-SI_GR5_ ← S-S Ind (GR5)					0.14 **	0.03	0.19 **	0.03
PIQ-SI_GR6_ ← S-S Out (GR6)					0.11 **	0.03	0.17 **	0.03
PIQ-SI_GR6_ ← O-S Ind (GR6)					0.23 **	0.03	0.16 **	0.03
PIQ-SI_GR6_ ← S-S Ind (GR6)					0.16 **	0.03	0.14 **	0.03
R^2^								
Δ PIQ-SI_GR6-GR5_	0.04 **	0.01	0.04 **	0.01	0.02 **	0.01	0.03 **	0.01
PIQ-SI_GR5_					0.02 *	0.01	0.08 **	0.02
PIQ-SI_GR6_					1 ^F^		1 ^F^	

Note. PIQ-SI = Social Inclusion subscale from Perceptions of Inclusion Questionnaire’; S-S = same sex; O-S = other sex; Out = out-degree; Ind = in-degree; GR = grade; Stability without O-S is not presented in the model because it serves as the reference category. ^b^ = parameters estimated based on the model invariant across time and in view of sex; ^F^ = fixed parameter. ** Statistical significance level (*p* < 0.01). * Statistical significance level (*p* < 0.05).

## Data Availability

The data presented in this study are available on request from the corresponding author.

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
