# Peer review of "The Inclusion of Other-Sex Peers in Peer Networks and Sense of Peer Integration in Early Adolescence: A Two-Wave Longitudinal Study"

_ijerph, 2022, doi:10.3390/ijerph192214971_

Round 1
Reviewer 1 Report
I have some minor comments in this paper. Please deal with my review comments.
0. Keywords
I suggest replacing "sociometry" with "sociometric popularity".
1. Introduction
I think this section is very broad. Please shorten it a bit, keeping the essence.
2. Methods
The authors were investigating peer integration, but I understand PIQ-SI was the indicator of satisfaction with peer relationships. Is peer integration have the same meaning as satisfaction with peer relations?
I think the authors can emphasize the difference between indegree and outdegree for the readers' easier understanding.
3. Result
In Figure3, the percent difference can be tested statistically.
I think that Other-Sex Outdegree was not used in LDS models without any reasoning. If so, please add the reason.
Figure 4 helped to understand the result of the analysis.
4. Discussion
Please clearly highlight the significance of the current study.
I hope these comments help.
Author Response
Response to Reviewer 1 Comments
We appreciate your comments. As discussed below, we have revised our manuscript with underlines based on your comments.
Reviewer’s comment 1:
I suggest replacing "sociometry" with "sociometric popularity".
Response:
As suggested, we have replaced "sociometry" with "sociometric popularity"
Reviewer’s comment 2:
I think this section is very broad. Please shorten it a bit, keeping the essence.
Response:
The introduction has been shortened. Passages only indirectly related to the research problem were cut out. We have removed, among other things, parts of the text: from line 33 to line 39; 41-44; 50-56; 64-70; 78-97; 101-106; 108-114; 155-156; 192-194; 230-234. The removal of passages shortened the Introduction by 30% (please look at the new version of the manuscript).
Reviewer’s comment 3:
The authors were investigating peer integration, but I understand PIQ-SI was the indicator of satisfaction with peer relationships. Is peer integration have the same meaning as satisfaction with peer relations?
Response:
Here, there has certainly been a misunderstanding. The PIQ-S is not an indicator of peer integration, but of a sense of peer integration and therefore a subjective perception of their quality. In the Introduction, we highlight the differences between objective peer relationships and a sense of the quality of these relationships (cf. lines 203-210). Throughout the manuscript, peer integration appears 11 times, but each time with the prefix "the sense of". For greater clarity, we have changed the title of the 2.1.3 paragraph. In the new version of the manuscript, it reads as follows:
“Measuring the (subjective) sense of peer integration (PIQ-SI)”.
Reviewer’s comment 4:
I think the authors can emphasize the difference between indegree and outdegree for the readers' easier understanding.
Response:
In the revised version of the manuscript, we have added the following passage (along with an additional Figure) in which we have tried to show the specificity of the two measures.
Let us assume that we have a network like the one presented in Panel A of Fig. 1. Student A nominates only student B. Students C nominate students A, B, and D. Students E nominates students A, but nobody nominates him/her. Panel B represents an adjacency matrix that contains the same information as the graph in panel A. From the lines, we can read who the group members have chosen and from the columns who was chosen by whom. For example, student B selected A (1 in column A). He himself was chosen by A (1 in row A of column B) and by C (1 in row C column B). Panel C shows the number of received and given nominations (column and row, respectively) and the values of in-degree and out-degree measures created by dividing the appropriate number of nominations by 4, that is, the number of group members minus 1 (see Figure 1).
As a result, In-degree centrality means the degree of relations that person X receives from others, and it’s based on the number of in-coming links. Student’s with high in-degree have more – compared to class peers – choices received from others, is more than others likeable. On the other hand, out-degree centrality, based on the number of outgoing links, means the degree of relations that person X sends toward others. In other words, the student with high out-degree is more active in choosing others, designates a lot – compared to other classmates – of other students as liked by him or her. In-degree and out-degree values range from 0 to 1. Coefficients were calculated in the igraph package [1].
Reviewer’s comment 5:
In Figure3, the percent difference can be tested statistically.
Response:
The authors thank the Reviewer for this comment. Following this comment, we would use three types of coefficients in our analyses: chi-square independence test, adjusted standardized residuals, and McNemar's test. We have added the results of these analyzes to the revised version of the manuscript. We have also added an additional paragraph (2.2.5 ‘Categorical data analysis’) to Section 2.2 ‘The Plan of Analyses’ with relevant explanations.
Analyzing the relationship between categorical variables, e.g. differences between boys and girls in the intensity of heterophilic preferences (in grades 5 and 6), we used a chi-square independence test [2]. For contingency tables larger than 2x2, adjusted standardized residuals (ASR) were also used [3]. These coefficients indicate whether its observed frequency is significantly different from the expected frequency. In consequence, ASR can be used to describe the pattern of association among the table cells and may be treated as a post hoc analysis of the chi-square (omnibus) test. Although the low level of statistical significance of the chi-square test shows that there are some significant differences in the contingency tables, the ASR indicates which cells are different from each other. As ASR has an asymptotic standard normal distribution, the absolute value greater than 1.96 indicates a deviation from independence in the cell. More specifically, if the ASR is less than -1.96, the observed frequency of the cell is lower than expected. Furthermore, if the ASR is greater than +1.96, the observed frequency is higher than expected [4].
To test the significance of differences intensity of heterophilic preferences between grade 5 and grade 6, we used McNemar's test [5]. McNemar's test is a chi-square test that is used to compare two proportions when the data are paired (that is, measured on the same sample). To distinguish between McNemar’s chi-square and regular chi-square independence tests, we used a prefix “M” denoting a McNemar’s test (e.g., Mχ2).
Reviewer’s comment 6:
I think that Other-Sex Outdegree was not used in LDS models without any reasoning. If so, please add the reason.
Response:
We thank the Reviewer for this important comment. In the modified version of the manuscript, we have added the following note:
It should be noted that the variable Other-Sex Outdegree (OS-Out) was not included in the regression model. This was because the information included in the OS-Out variable (student indicates someone of the other gender or does not indicate) was partly used to construct a categorical variable describing the cross-sex relationships occurring between 5th and 6th grade (Stability without O-S, Stability with O-S, Profit O-S and Loss O-S), a key predictor of changes in the sense of quality of peer relationships (cf. Figure 3). Since both variables share overlap information, including OS-Out in the regression model could result in biased estimation of key independent effects predictor variables on the outcome variable [6].
Reviewer’s comment 7:
Please clearly highlight the significance of the current study.
Response:
In the modified version of the manuscript, we have added the following passage in the Conclusions paragraph:
However, the main objective of the present paper was to determine the relationship between changes in heterophily and changes in perception of the quality of peer relationships. To the best of the authors' knowledge to date, there have been no empirical studies based on longitudinal data that would determine if, and if so in what way, the process of including the other sex in originally homophilic social circles is linked to the dynamics of the perceived quality of peer relationships. In the cross-sectional perspective, correlational analyses showed that both liking other-sex peers (Other-Sex Outdegree) and being liked by them (Other-Sex Indegree) are associated with better perceptions of peer relationships, as are liking same-sex peers (Same-Sex Outdegree) and being liked by them (Other-Sex-Indegree).
Importantly, our research adds to the previous knowledge empirical evidence from a longitudinal study that the process of inclusion (between the 5th and 6th grades) of other-sex peers in initially same-sex peer relationships translates into a better perception of the quality of peer relationships (both among boys and girls). The results additionally indicate that in the case of boys, the loss of relationships with girls has a negative impact on the sense of peer integration. In this way, our findings support the hypothesis [7] that the process of establishing relationships with the other-sex peers serves an important developmental function by expanding the pool of attachment figures available to meet the social and emotional needs and that this process may improve not only the perceived quality of peer relationships, but also broader psychosocial well-being, especially for the early adolescents with a deficit in positive same-sex peer relationships.
References
- Csardi, G.; Napusz, T. The Igraph Software Package for Complex Network Research. InterJournal 2006, Complex Systems, 1695.
- Plackett, R.L. Karl Pearson and the Chi-Squared Test. Int. Stat. Rev. Rev. Int. Stat. 1983, 51, 59, doi:10.2307/1402731.
- Agresti, A. Categorical Data Analysis; Wiley series in probability and statistics; 2nd ed.; Wiley-Interscience: New York, 2002; ISBN 978-0-471-36093-3.
- Field, A. Discovering Statistics Using IBM SPSS Statistics; 5th edition.; SAGE Publications: Thousand Oaks, CA, 2017; ISBN 978-1-5264-1952-1.
- McNemar, Q. Note on the Sampling Error of the Difference between Correlated Proportions or Percentages. Psychometrika 1947, 12, 153–157, doi:10.1007/BF02295996.
- Vatcheva, K.P.; Lee, M. Multicollinearity in Regression Analyses Conducted in Epidemiologic Studies. Epidemiol. Open Access 2016, 06, doi:10.4172/2161-1165.1000227.
- Sippola, L.K. Getting to Know the “Other”: The Characteristics and Developmental Significance of Other-Sex Relationships in Adolescence. J. Youth Adolesc. 1999, 28, 407–418, doi:10.1023/A:1021660823003.

Reviewer 2 Report
It was my pleasure to review this manuscript about peer relationships in early adolescence in Poland. The findings of the study are important and give us a lot to think about. I have just a few minor comments:
Introduction:
- The chapter provides lots of useful information, but is quite long. Could it be shortened in places?
Conclusions:
- The chapter is missing. Could you please provide a short summary of your findings?
Institutional Review Board Statement, Informed Consent Statement – missing
Author Response
Response to Reviewer 1 Comments
We appreciate your comments and suggestions and after rethinking them we give the answers and add some passages to the text.
Reviewer’s comment 1:
The chapter provides lots of useful information, but is quite long. Could it be shortened in places?
Response:
The introduction has been shortened. Passages only indirectly related to the research problem were cut out. We have removed, among other things, parts of the text: from line 33 to line 39; 41-44; 50-56; 64-70; 78-97; 101-106; 108-114; 155-156; 192-194; 230-234. The removal of passages shortened the Introduction by 30% (please look at the new version of the manuscript).
Reviewer’s comment 2:
The chapter is missing. Could you please provide a short summary of your findings?
Response:
Thank you for this comment. In the new version of the manuscript, we have added the section Conclusions
Conclusions
It is a well-known fact that in adolescence, the importance of peer relationships increases significantly, and the sense of peer integration is associated with solving developmental tasks and psychological and social well-being. Previous studies confirm that in the stage of early adolescence, in peer interactions choosing same-sex friends is the dominant trend although it is beginning to be accompanied by the appearance of cross-sex choices (sex-related heterophily within peer relations), which are a preliminary stage for the development of romantic relationships. In general, the results of the present study converge with previous theory and research indicating that the level of heterophily (choosing peers of the other sex as liked) is higher – in grades 5 and 6 – among girls than among boys. Our study also confirmed that between the ages of 11 and 13, the intensity of heterophily increases significantly (by approximately 10 percentage points). While in grade 5 only 39.5% of the boys liked at least one person of the other sex, in grade 6 this value was already 49.4%. For girls, on the other hand, the level of heterophily increased from 47.5% to 56.2%.
However, the main objective of the present paper was to determine the relationship between changes in heterophily and changes in perception of the quality of peer relationships. To the best of the authors' knowledge to date, there have been no empirical studies based on longitudinal data that would determine if, and if so in what way, the process of including the other sex in originally homophilic social circles is linked to the dynamics of the perceived quality of peer relationships. In the cross-sectional perspective, correlational analyses showed that both liking other-sex peers (Other-Sex Outdegree) and being liked by them (Other-Sex Indegree) are associated with better perceptions of peer relationships, as are liking same-sex peers (Same-Sex Outdegree) and being liked by them (Other-Sex Indegree).
Importantly, our research adds to the previous knowledge empirical evidence from a longitudinal study that the process of inclusion (between the 5th and 6th grades) of other-sex peers in initially same-sex peer relationships translates into a better perception of the quality of peer relationships (both among boys and girls). The results additionally indicate that in the case of boys, the loss of relationships with girls has a negative impact on the sense of peer integration. In this way, our findings support the hypothesis [1] that the process of establishing relationships with the other-sex peers serves an important developmental function by expanding the pool of attachment figures available to meet the social and emotional needs and that this process may improve not only the perceived quality of peer relationships, but also broader psychosocial well-being, especially for the early adolescents with a deficit in positive same-sex peer relationships.
Reviewer’s comment 3:
Institutional Review Board Statement, Informed Consent Statement – missing
Response:
In the modified version of the manuscript, we have added both missing information:
Institutional Review Board Statement
The study was conducted according to the Declaration of Helsinki and approved by the Scientific Research Board of the Educational Research Institute in Warsaw (Instytut Badań Edukacyjnych w Warszawie).
Informed Consent Statement
Informed consent was obtained from all subjects involved in the study.
References
- Sippola, L.K. Getting to Know the “Other”: The Characteristics and Developmental Significance of Other-Sex Relationships in Adolescence. J. Youth Adolesc. 1999, 28, 407–418, doi:10.1023/A:1021660823003.
Round 2
Reviewer 1 Report
The authors have addressed my comments and I think the manuscript can be accepted.